# The Impact of Four High-Altitude Training Camps on the Aerobic Capacity of a Short Track PyeongChang 2018 Olympian: A Case Study

**DOI:** 10.3390/ijerph19073814

**Published:** 2022-03-23

**Authors:** Anna Lukanova-Jakubowska, Katarzyna Piechota, Tomasz Grzywacz, Tadeusz Ambroży, Łukasz Rydzik, Mariusz Ozimek

**Affiliations:** 1Faculty of Physical Education and Physiotherapy, Opole University of Technology, Próskowska 76, 45-758 Opole, Poland; ajakubowska2@wp.pl; 2Department of Sport, Institute of Physical Culture, Kazimierz Wielki University in Bydgoszcz, Chodkiewicza 30, 85-064 Bydgoszcz, Poland; tomasz.grzywacz@sport-olimpijski.pl; 3Institute of Sport Science, University of Physical Education in Krakow, Al. Jana Pawła II 78, 31-541 Kraków, Poland; tadek@ambrozy.pl (T.A.); mariusz.ozimek@awf.krakow.pl (M.O.)

**Keywords:** high-altitude training, hypobaric hypoxia, short-track, CPET graded exercise test, Wingate anaerobic test, Olympic athlete

## Abstract

This study characterizes high-altitude training camps and their effect on the aerobic capacity of a Polish national team member (M.W.), who was a participant in the PyeongChang 2018 Winter Olympic Games (body weight: 59.6 kg, body height: 161.0 cm, fat mass: 10.9 kg and 18.3% of fat tissue, fat-free mass: 48.7 kg, muscle mass: 46.3 kg, and BMI = 23.0 kg/m^2^). The tests were conducted in the periods from April 2018 to September 2018 and April 2019 to September 2019 (period of general and special preparation). The study evaluated aerobic and anaerobic capacity determined by laboratory tests, a cardiopulmonary graded exercise test to exhaustion performed on a cycle ergometer (CPET), and the Wingate anaerobic test. Based on the research, training in hypobaric conditions translated into significant improvements in the skater’s exercise capacity recorded after participating in the Olympic Winter Games in Korea (February 2018). In the analyzed period (2018–2019), there was a significant increase in key parameters of aerobic fitness such as anaerobic threshold power output (AT-PO) [W]—223; power output POmax [W]—299 and AT-PO [W/kg]—3.50; (POmax) [W/kg]—4.69; and AT-VO_2_ [mL/kg/min]—51.3; VO_2_max [mL/kg/min]—61.0. The athlete showed high-exercise-induced adaptations and improvements in the aerobic metabolic potential after two seasons, in which four training camps were held in altitude conditions.

## 1. Introduction

High-altitude training is based on the fact that with increasing altitude (above sea level), the atmospheric pressure decreases and, consequently, partial pressure of oxygen in the inspiratory air also decreases. The flattening of the cascade of oxygen transport from the lungs to the tissues leads to a decrease in blood oxygen saturation and causes oxygen deficiency in peripheral tissues [1].

Short-track speed skating has been an Olympic sport since 1992, and it first appeared in the program of the Albertville Winter Olympics. Since then, it has been dynamically developing and gaining recognition in the world arenas, especially in Asian countries such as Japan, Korea, or China, but also in the USA and Canada [2]. Short track is a winter sport that combines two aspects of motor skills: speed and endurance. In addition to perfect technique, an important role is also played by the tactical preparation of both skaters and coaches [3,4,5]. Therefore, competition in short-track events is as spectacular as it is very risky and dangerous. 

This, in turn, is associated with the body’s defensive responses (increase in minute ventilation of the lungs and the heart rate), which forces it to be much more functional during physical exertion [6]. Increasing the effort of the body is a natural mechanism aimed to prevent hypoxia in the body’s systems, organs, or tissues, and the deterioration of their functions [7]. During training in high altitude conditions and with increased load, the body experiences fatigue faster, which increases the effect of the training stimulus, stimulates the compensatory mechanisms more efficiently, and consequently leads to adaptive changes conducive to better oxygen transport and improvement in physical efficiency [6,8]. Paradoxically, one of the effects of high-altitude training is the reduction in oxygen demand (economizing motor activities) and a decrease in the ability to maximize acidification, enabling extension of the working time during a training session [9,10,11].

Studies of many authors have shown that the training process conducted in conditions of altitude hypoxia is very effective for athletes practicing endurance and mixed sports [12,13,14,15]. Training in conditions of hypobaric hypoxia causes many physiological changes in the body. They are very similar to the changes caused by endurance training. The most common increases are observed in lung ventilation parameters, blood hematological indicators (the number of erythrocytes, hematocrit, and hemoglobin level), the density of capillaries, the amount and volume of mitochondria, and the activity of oxidative enzymes in muscles [16,17]. In the research of Biggs et al. [18], similar effects were obtained with simulated hypoxia induced artificially by breathing through gas masks (with hypoxic air), where an increase in some physiological parameters was observed, including the most important one—VO_2_max. The three basic strategies for training in hypoxia are (1) live-high–train-low, in which normobaric hypoxia conditions simulate a 2000–3000 m altitude with 15.3% oxygen concentration and increased nitrogen concentration where the athlete lives, but training sessions are performed in low-altitude conditions [19]; (2) live-high–train-high, which consists of living and training at 2000–3000 m above sea level [20]; and (3) live-low–train-high [21].

High altitudes above sea level significantly reduce the maximum oxygen uptake (VO_2_max). It is believed that, at altitudes up to 1500 m above sea level, the VO_2_max level does not decrease significantly and does not limit exercise capacity. However, at altitudes over 1500 m above sea level, there is a significant decrease in VO_2_max by approx. 11% per each 1000 m [22]. Training under hypoxic conditions stimulates the economics of oxygen transport to and in the working muscles. During training in alpine terrain, the body acquires more and more functional skills to uptake and absorb oxygen available in the mountain air. This modifies the mechanism related to oxygen transport in working muscles, which is one of the key factors influencing the level of VO_2_max. It takes about 14 days of high-altitude training to improve anaerobic capacity. On the other hand, the improvement of aerobic capacity (VO_2_max level) takes much longer, as it requires a permanent stay at an altitude of about 2000–3500 m above sea level for a period of 3–4 weeks and longer [1]. 

The optimal action would be to conduct training of elite athletes under conditions of high-altitude hypoxia in all periods of sports preparation. However, in short track, due to the specificity of the sport and the insufficient availability of ice tracks located at altitudes of over 1800 m above sea level, training is performed mainly during the general preparation period [13].

In order to constantly improve the training process, it is extremely important to constantly search and implement new but proven methods. One of such solutions is certainly the use of rational diagnostic solutions to obtain additional information about the functional efficiency of the athlete’s body. This is especially important in competitive athletes, where any change, even the smallest, can contribute to the improvement of sports performance. Constant observation of the professional profile and cyclical performance of tests are aimed at supporting the optimization and individualization of the training process and lead to high performance at sports championships. Such research results may lead to a more rational training process and allow for the assessment of the usefulness of training camps in high-altitude areas on the level of fitness of short-track athletes in the annual training cycle of preparation for championships. 

The aim of this study is to attempt to characterize the training process in high-altitude conditions and its impact on the physical fitness of a Polish short-track team member highly ranked at the international level. The research hypothesis assumes that training in high mountain conditions will significantly affect the physical fitness of the athlete. 

## 2. Materials and Methods

### 2.1. Subject

M.W., a Polish speed skater specializing in short track, participant of the Winter Olympic Games in PyeongChang 2018 (body weight: 59.6 kg, body height: 161.0 cm; fat mass: 10.9 kg and 18.3% of fat tissue; fat-free mass: 48.7 kg, muscle mass: 46.3 kg, and BMI = 23.0 kg/m^2^). At that time, the 21-year-old competitor obtained Olympic qualification for three events: 500 m, 1000 m, and 1500 m. The leading distance was 1000 m, where she won 12th place in the Olympic Games. She has been a member of the AZS University Club of the Opole University of Technology since 2015. In the Olympic season, the athlete obtained the best results in the multistage test at the level of: AT-PO (W/kg)—3.59; and AT-VO_2_ (mL/kg/min)—45.7. Maximum values: POmax (W/kg)—4.77; and VO_2_max (mL/kg/min)—55.8 [23].

### 2.2. Methods and Tools

The research was carried out during two annual training cycles after the PyeongChang 2018 Winter Olympics. Four training camps were held in high-altitude conditions, mainly during the general and special preparation period, which lasted from April 2018 to September 2019 (test dates in high mountain conditions: 21 September 2018, 6 May 2019, 10 June 2019, and 8 August 2019). In total, before and during the training camps under hypoxic conditions, the research was conducted seven times. The standard incremental-intensity, cardiopulmonary exercise test on Monark 874 E cycle ergometer (CPET) (Sweden) and Wingate anaerobic test were used. The main method used in the research was the case study [24,25,26,27], maintaining its basic assumptions but with its own modifications by adjusting it to the purpose and needs of our research. These modifications concerned the reference of the method to the study of an individual sports career, the development of results, and opinions on training and non-training factors of the short-track athlete—M.W.

### 2.3. Research Procedures

The tests were carried out at the Institute of Sport—National Research Institute in Warsaw (Poland). The athlete tested was informed about the purpose and measurement method and the possibility of withdrawal from participation in the study at any time without giving reasons. She also gave her written informed consent to participate in the study.

Before starting the tests, the athlete was medically qualified to perform the test exercise, and each time before and after the stress test, a computerized 12-lead ECG (BTL 08MT ECG No. 1230, BTL Industries Limited, Stevenage, Hertfordshire, UK) was also performed. 

Cycle ergometer CPET and Wingate test were performed 4 times in 2018 and 3 times in 2019. 

The athlete performed a 30 s Wingate anaerobic test for the lower limbs with a relative load of 7.5% of body weight. The test was performed on a Monark 874 E cycle ergometer (Sweden). The test was preceded by an individual warm-up. Previously, the Wingate test was used by leading skaters and speed skaters as a diagnostic measure to determine the take-off ability over a distance of 1500 m [28]. 

Two hours after the Wingate test, aerobic fitness was evaluated using a graded exercise test to exhaustion. The exercise test was preceded by a 10 min warm-up on a cycle ergometer (stationary) at a load of 50 W. The test consisted of 3 min stages, not separated by a break, with each subsequent stage performed at an increased load. The load was adjusted individually to the athlete’s body weight: it was 1.0 W/kg body weight in the first stage, and in each subsequent stage, it was increased by 0.7 W/kg body weight.

Respiratory indicators such as minute ventilation, oxygen uptake, and carbon dioxide excretion were measured by the “breath by breath” method using a Cortex MetaMax 3B ergospirometer (Biophysik GmbH, Leipzig, Germany) with a 15 s average. Previous studies [29,30] confirmed that this measurement system is reliable for oxygen uptake measurements. Before each test, a two-step calibration of the respiratory gas measurement system was performed. Ambient gas calibration was performed, and a correction factor was set to the gas calibration values (14.97% O_2_, 4.96% CO_2_, balance N_2_: ±0.02% absolute, Hong Kong Specialty Gases) and volume calibration with a 3 L calibration pump 5530 series (Hans Rudolph, Inc., Shawnee, KS, USA). Staged test measurements were made on an Excalibur Sport cycle ergometer (Lode, Groningen, The Netherlands). 

Lactate levels were evaluated in capillary blood taken from the fingertip immediately and 3 min after the completion of the test using Dr. Mueller’s Super GL2 analyzer (Dr. Müller Gerätebau GmbH, Freitel, Germany). 

Based on the graded exercise test, the anaerobic threshold (AT4) was determined by interpolation for the blood lactate concentration of 4 mmol/L.

During the test, heart rate (HR) was recorded using a Polar system (Polar Electro Oy, Kempele, Finland). 

The research was approved by the Scientific Research Ethics Committee of the Institute of Sport—National Research Institute.

### 2.4. High-Altitude Training of an Olympic Athlete

After the analysis of the sports skill level of the skater after the 2018 Olympic Games, it was found that the main training goal for the next two years will be to increase the level of anaerobic capacity, especially at the anaerobic threshold AT (VO_2_, PO [W], PO [W/kg]), and the maximum values. It was assumed that increasing these parameters would improve the performance in 1000 m and 1500 m events.

From a physiological point of view, a higher VO_2_max level would increase the skating economy during the first laps of the distance, which, in turn, would translate into the possibility of using the skater’s power at the end of the distance. Short-track skaters must control their exercise over the distance in such a way that they achieve the highest speed at the end of the distance (attacking or overtaking opponents). 

The athlete’s basic competitive distance was 1000 m. It requires an effort lasting about 80–81 s. From the physiological point of view, it is exercise at the third level on the intensity scale according to bioenergy criteria [31,32,33], which is a typically anaerobic lactic acid work. The most important ability of an athlete in such conditions is tolerance to acidification while doing as much work as possible (and developing maximum speed). However, in order to be able to obtain the training effect in the form of improved lactic acid tolerance, an important factor is the high efficiency of oxygen utilization. In the training process, high VO_2_max supports faster post-exercise recovery, especially after anaerobic exercise (characterized by high levels of lactic acid). 

In the 2018 Olympic season, the athlete’s best time at 1000 m was 1:30.24 [23]. This ensured her 12th place at the 2018 Olympic Games. In order to be able to win the medals of the World Championships and then of the Olympic Games, another important factor was the introduction of additional stimuli to the training process in the form of camps in high altitude conditions. The idea was to achieve a higher VO_2_max level and to do more total work. This was expected to translate into greater lactic acid tolerance and better sports performance. The alpine training camps were held in Erzurum, Turkey, at an altitude of 1950 m above sea level and in Ankara, Turkey, at an altitude of 938 m above sea level (Table 1). The training conditions in Erzurum were conducive to the execution of a standard training plan (ice track, treadmill, gym). Additional training stimuli were used in the form of various running forms and cycling in the mountains at an altitude of 2200 m above sea level.

During all training camps, the main load was specialized training on ice. Between the training camps, the skater also participated in training sessions in the AZS PO Opole club. High-altitude training was performed four times, cyclically over an 18-month period. From the point of view of the theory of sport, this period falls within the scope of two periods of general and special preparation in the two consecutive competitive seasons 2018/2019 and 2019/2020 [33]. 

The four individual mesocycles were divided into 4 microcycles each. The microcycles were planned in the following sequence: introductory–build-up–build-up–regenerating. The number of training sessions in the introductory and build-up microcycles was 11, and in the regeneration microcycle, it was 6. The daily training duration was 4 to 6 h, respectively, and 1–3 h in the regenerative microcycle. The training sessions took place on the ice, an athletics track, gym, on a bicycle, and in the field (high altitude of over 2100 m above sea level).

A detailed description of the training process in high-altitude conditions for an Olympic athlete (M.W.) in the 2018/2019 and 2019/2020 seasons:

Table 2 below shows the training means used during the individual mesocycles whereas Table 3 and Table 4 show the number of training sessions performed.
1In macrocycles of general preparation, the main training was based on aerobic training—continuous method (efforts at threshold intensity of AT4) and interval lactic acid training.
-The combination of the two workouts was considered to increase VO_2_max. The training sessions were performed on a bicycle, measuring work at a specific heart rate (HR). Training was also focused on running or walking in the mountains. Duration of training sessions: 60–120 min.-Lactic acid training sessions were held two or three times in a microcycle using the interval method. The training consisted of running on the ice or running a certain number of sections on an athletics track in 50″–1′45″ at a speed of 85% with a break between sections 1′–2′. The number of repetitions in a set was 7–10, with 2–3 sets.2The training intensity changed to 90% in the second mesocycle. This intensity resulted in better production of lactic acid in the muscles and an increase in the athlete’s blood lactate level. On this basis, the plan includes typical aerobic training conducted on a bicycle and in a running form. They supported faster regeneration of the body after exercise.
-An important assumption at that time was the performance of endurance tests, especially the graded exercise test, which allowed us to determine the athlete’s HR at the metabolic thresholds, enabling better programming of subsequent training sessions and choice of their intensity (LT and AT).-Complementary training: aerobic training was performed only in specific zones based on the principle of training individualization. For lactic acid tolerance training to be effective and not to lead to overtraining or fatigue after glycogen depletion, aerobic training in specific AT and LT zones was very important, thus leading to improved performance.-The aerobic training was performed for 4–5 sessions in a given build-up microcycle, and up to 6–7 sessions in the introductory microcycle.
3Strength training also played an important role in the training plan. It was performed 3 times in the build-up microcycle and twice in the introductory microcycle.-In the regenerative microcycle, one training session was performed for muscle stimulation. In the preparatory period, strength training was focused on strength endurance and building maximum strength. During the competitive period, strength training was aimed at maintaining maximum strength and building maximum strength. The range of training loads was controlled and adjusted every four weeks.4Complementary training was ice speed training, ice skating technique training, and skating tactics training. These training sessions were performed using the individual skating method or in a relay race. Such training sessions were designed to combine main training with support training.5In the regenerative microcycle, an important factor was the work with a physiotherapist and complementary training for the athlete.

## 3. Results

The results of this study are presented divided into four mesocycles, started after the Winter Olympics. In the beginning, the assumptions were made before the season by the coach, and then the goals achieved in a given season were discussed.

Table 5 below shows the results of diagnostic tests performed during the period of general and special preparation in the 2018/2019 and 2019/2020 seasons (in May–August).

After the Olympic season, the athlete had a break from training on 1–20 April 2018. The test results (on 29 May 2018) showed a change in all parameters in the graded exercise test, both at the anaerobic threshold (AT) and in the maximum values (Table 5 and Table 6).

On 20–27 May 2018, preparations for the season began, which took place in Opole. The main goal was to gradually adapt the athlete to the exercise and increase its intensity. In the period from May to July, training sessions were held in the field, and in August, additionally on ice. During this period, two control tests were carried out, resulting from the need to evaluate individual HR at AT levels to determine the zones of aerobic training intensity. Table 5 shows that the graded exercise test results were gradually improving, especially at AT. 

The research conducted after the first alpine training camp (21 September 2018) showed a significant improvement in the parameters in the graded exercise test (Table 6). Particularly noteworthy are the values at the level of maximum effort, which were, respectively: maximum power POmax (W)—298; VO_2_max (L/min)—3.80; and VO_2_max (mL/kg/min)—60.3.

It turned out that the athlete’s indices were at the highest level since the beginning of her sports career. The maximum oxygen uptake VO_2_max of 60 mL/kg/min was exceeded for the first time. 

After the end of the 2018/2019 competitive season, control tests were carried out before the first alpine training camp in the second season (6 May 2019). The aim of the diagnostic tests was to determine the level of the athlete’s exercise capacity and to determine the heart rate at AT after the detraining break, which lasted 3 weeks. The determination of these data allowed for the development of a cycling and aerobic training program. It is noteworthy that the values at AT improved, but not as significantly as the maximum levels, which, so far, had been the best for the athlete since the beginning of her sports career.

Further tests were carried out after the end of the second alpine training camp (8 August 2019) in the 2019/2020 season. At this stage of training, the athlete achieved the best results in the graded test, both at the AT level and in the maximum values. Compared to the athlete’s results before the Olympic Games (Table 5), the results achieved in the tests during this period were definitely at a higher level, which amounted to, respectively:-AT-VO_2_ (mL/kg/min)—51.3; VO_2_max (mL/kg/min)—61.0;-AT-PO (W)—223; POmax (W)—299;-AT-PO (W/kg)—3.50, POmax (W/kg)—4.69.

The lower values of the parameters relative to kg of the athlete’s body weight resulted from the higher body weight of the skater compared to the periods of competitions in the Olympic Games. In this study, the lactate concentration curve (LA [mmol/L] in the graded exercise test (Figure 1) clearly shows the most favorable course. The obtained data confirm high threshold levels, which indicates high exercise-induced adaptations to aerobic loads. The LA–power dependence curve clearly shifts to the right, which means that the same values of lactate levels occur at increasingly higher power.

In the last two tests, maximum power and maximum oxygen uptake reach significantly higher levels. This picture of changes confirms the more economic work in the zone of higher exercise intensity, with a greater contribution of aerobic processes. This demonstrates the skater’s excellent aerobic potential.

Trend analysis for physiological variables (PV) examined in four time points during preparatory training 2018 is presented in Table 6.

The obesity of high correlations in AT4 was demonstrated, which are detailed in Table 7.

## 4. Discussion

High-altitude training induces the state of hypoxia in the body as a result of lowering blood oxygen saturation, defined as high-altitude hypoxia, or hypoxic hypoxia. Many scientific sources provide information that alpine training in the period of general preparation of athletes has a beneficial effect on the body of athletes practicing endurance sports [12,13]. These studies are in line with the studies of other authors, which confirm that training conducted in the conditions of alpine hypoxia (especially during general preparation) has a positive effect on adaptive changes and the values of physiological parameters (at AT and in conditions of maximum exercise). 

Effective technical and tactical preparation and high motor skills are essential for short-track skaters. Due to the varying distances (500 m, 1000 m, and 1500 m), skaters practice in all ranges of exercise intensity. Their main training measures include zone II (aerobic, ca. 35%) of training loads and III (mixed exercise, ca. 38%) [29]. However, as the coach of the Polish IO team member emphasizes, anaerobic metabolism is important, especially at 500 m and 1000 m [4,31]. On the other hand, skating the distance of 1500 m does not only involve anaerobic exercise, but, in 50% of cases, the aerobic system (intensity zone IV: between 2–3 min of exercise) falls within the second intensity range according to bioenergy criteria, i.e., 50% to 50% [23,34]. Even at longer distances, e.g., 1500 or 3000 m, the athlete must also show speed abilities, i.e., sprint skills [28]. Therefore, short-track athletes use training stimuli from all exercise intensity zones. Based on possible ways to increase these abilities, coaches are constantly looking for new effective training solutions. 

The research conducted after the first alpine training camp (21 September 2018) showed a significant improvement in the parameters in the graded test. Particularly noteworthy are the values at the level of maximum effort, such as maximum power POmax (W): 298; VO_2_max (L/min): 3.80; and VO_2_max (mL/kg/min): 60.3. It turned out that the athlete’s indices were at the highest level since the beginning of her sports career. The oxygen uptake of VO_2_max—60 mL/kg/min was exceeded for the first time. This demonstrates the progressiveness induced by training under hypoxia. Similar findings were presented by Czuba et al., who claimed that hypoxia training is an effective means of training to improve aerobic capacity [35]. In contrast, Morton and Cable found that acute exposure of moderately trained individuals to normobaric hypoxia during a short-term training program consisting of intermittent moderate- to high-intensity exercise has no increased effect on the degree of improvement in aerobic or anaerobic capacity [36]. The present analysis concerns an athlete presenting at the elite performance level who was administered the exposure to prolonged hypoxia. Therefore, the improvement in results may be related to the length of the training cycle in high-altitude conditions. 

After the end of the 2018/2019 competition season, control tests were carried out before the first alpine training camp in the second season (6 May 2019). The aim of the diagnostic tests was to determine the level of exercise capacity and to determine the heart rate at AT after the detraining break (3 weeks). The determination of these data allowed the development of a cycling and aerobic training program. 

Further tests were performed after the end of the second alpine training camp (8 August 2019) in the 2018/2019 season. At this stage of training, the athlete achieved the best results in the graded exercise test, both at the AT level and in the maximum values: AT-VO_2_ (mL/kg/min)—51.3; VO_2_max (mL/kg/min)—61.0; AT-PO (W)—223; and POmax (W)—299.

In the tests of lactate levels (LA) (mmol/L) in the graded exercise test, noticeably high power thresholds were observed, which indicates exercise adaptations to the aerobic load. This confirms the more economic work of the athlete, performed at a greater contribution of aerobic processes and work in the zone of higher exercise intensity. This demonstrates the skater’s very high aerobic potential. 

## 5. Conclusions

Based on the conducted research that consisted of high-altitude training in the form of four camps under conditions of hypobaric hypoxia, the following practical conclusions can be drawn:1The use of four altitude camps in two consecutive annual training cycles allowed the athlete to achieve high exercise adaptations and improve the aerobic potential.2Significant improvements in physiological parameters were found, some of them reaching the highest levels in the professional skater’s career so far: threshold of anaerobic changes AT-VO_2_ (mL/kg/min) of 51.3, and VO_2_max (mL/kg/min) of 61.0.3The results of the study show unequivocally the high economy and excellent aerobic potential of the athlete.

## Figures and Tables

**Figure 1 ijerph-19-03814-f001:**
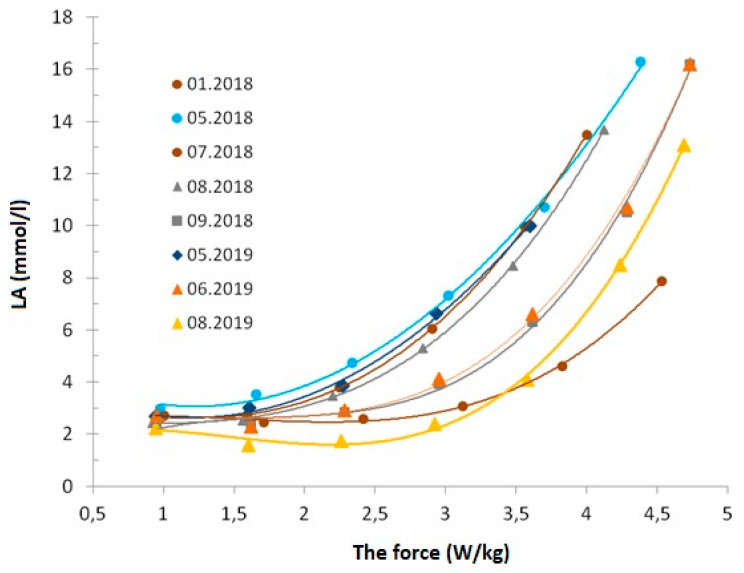
LA level (mmol/L) and maximum aerobic power obtained in the graded exercise test performed by the athlete in the periods studied.

**Table 1 ijerph-19-03814-t001:** Schedule of training camps in high altitude conditions planned for the athlete (M.W.).

Lp.	Camps Date	Place
1.	27.08–17.09.2018	Erzurum
2.	12.05–02.06.2019	Erzurum
3.	07.07–28.07.2019	Erzurum
4.	01.09–22.09.2019	Ankara

**Table 2 ijerph-19-03814-t002:** Examples of training measures (strength and power training) used in the microcycle of preparation of a short-track athlete.

Lp.	Training Measures—Microcycle	Load
1	Strength training - squat with a barbell (main exercise)	2 × 6—70% max 2 × 4—85% max 2 × 2—90% max 1 × 1—100% max
2	Power training - barbell jumping - jumping (athletics hurdles, stairs, other platforms) - sprint with various loads, medicine ball throws	4 × 10 barbell jumps—from half a squat 4 × 6 lunges—from a full squat

**Table 3 ijerph-19-03814-t003:** Dry land training in individual training periods.

Period/Numberof Training Microcycles	Test Date	I—Dry Land Training/HR/			II—Dry Land Training /Number of Training Sessions/			III—Dry Land Training /Number of Training Sessions/			IV—Dry Land Training /Number of Training Sessions/		
		LT—Up to 2 mmol	LT-AT—2–4 mmol	AT and Above	LT—Up to 2 mmol	LT-AT 2–4 mmol	AT and Above	Intensity			Intensity		
								85%	90%	95–100%	65–80%	80–95%	95–100%
I general 8 microcycles	29.05.2018	140	150	163	41	24	12	7	2	6	24	8	2
I special 8 microcycles	23.07.2018	150	160	171–176	36	18	16	2	4	5	6	12	10
II general 8 microcycles	06.05.2019	150	160	177	34	18	16	9	4	8	26	12	2
II special 8 microcycles	08.08.2019	170	180	191	28	18	16	4	6	6	6	12	12
*p*-value	I vs. II *p* < 0.001							III vs. IV *p* = 0.04					

Notes: I—dry land training—bike (aerobic and anaerobic range), 60–120 min—duration of the training session, HR—heart rate; II—dry land training—bike (aerobic and anaerobic range, 60–120 min—duration of the training session; III—dry land training—interval running at the distances 250 m, 500 m—anaerobic work; IV—on-ice training: relay training, training focused on distances of 1500 m, 1000 m, 500 m, LT, AT—aerobic and anaerobic threshold values.

**Table 4 ijerph-19-03814-t004:** Strength training in particular training periods.

Period/Number of Training Microcycles	Test Date	I—Strength Training/Number of Training Sessions			II—Specific Strength/Number of Training Sessions	
		Training Measures				
		Strength Endurance (Deadlift, Squat, Bench Press, Throw), 12–15 Repetitions 3–4 Sets	Maximum Strength (Deadlift, Squat, Bench Press, Chargé) 2–6 Repetitions 2–4 Sets	Maximum Power (Barbell Jumps, Charges, Plyometric Exercises), 4–6 Repetitions 3–4 Sets	Skating Imitations with Elements of Statics	Specialized Exercises on Belts and Rubbers
I general 8 microcycles	29.05.2018	22	4	2	24	26
I special 8 microcycles	23.07.2018	4	16	12	10	12
II general 8 microcycles	06.05.2019	22	4	2	22	26
II special 8 microcycles	08.08.2019	4	16	12	12	12

Notes: I—Strength training/number of training sessions; II—Specific strength/number of training sessions.

**Table 5 ijerph-19-03814-t005:** Results of the graded exercise test on a cycle ergometer recorded for a short-track skater in the 2018/2019 and 2019/2020 seasons (general preparation mesocycle, special preparation mesocycle).

	Date	LA [mmol/L]	PO [W]	PO [W/kg]	VO_2_ [L/min]	VO_2_ [mL/kg/min]	HR [sk/min]
**AT4**	29-05-2018	4.0	121	1.96	1.69	27.4	163
	23-07-2018	4.0	150	2.34	2.16	33.8	171
	22-08-2018	4.0	161	2.45	2.14	32.7	176
	21-09-2018	4.0	188	2.98	2.70	42.8	182
	06-05-2019	4.0	142	2.24	2.01	31.7	177
	10-06-2019	4.0	183	2.90	2.56	40.6	175
	08-08-2019	4.0	223	3.50	3.27	51.3	191
**MAX**	29-05-2018	16.4	270	4.43	3.26	52.9	198
	23-07-2018	13.5	256	4.00	3.25	50.8	195
	22-08-2018	15.1	284	4.34	3.35	51.1	203
	21-09-2018	16.2	298	4.73	3.80	60.3	206
	06-05-2019	13.8	242	3.82	3.06	48.3	202
	10-06-2019	16.2	298	4.73	3.83	60.7	201
	08-08-2019	13.1	299	4.69	3.88	61.0	208

Explanation of abbreviations: LA—blood lactate concentration, PO—relative power output, VO_2_—oxygen uptake, VO_2_max—maximum oxygen uptake, HR—heart rate, AT4—anaerobic threshold for blood lactate concentration 4 mmol/L.

**Table 6 ijerph-19-03814-t006:** Trends’ regressions expressed as the formula PV = S × day of preparation.

Effort	PV	Slope (S)	t-Value	*p*-Value
AT4	HR (b/min)	1.949	2.83	0.066
	PO (W/kg)	0.029	**3.6**	**0.037**
	VO_2_ (mL/kg/min)	0.411	**3.6**	**0.037**
MAX	HR	2.22	2.68	0.075
	PO	0.049	2.71	0.073
	VO_2_	0.506	**2.81**	**0.047**
	LA mmol/L	1.67	2.54	0.084

Values in bold are statistically.

**Table 7 ijerph-19-03814-t007:** Simple correlation coefficients between variables. Pooled data from 2018 and 2018 (*n* = 6).

Effort	Correlations	r-Value
AT4	HR ● PO	**0.802**
	HR ● VO_2_	**0.837**
	PO ● VO_2_	**0.938**
MAX	HR ● PO	0.441
	HR ● VO_2_	0.420
	PO ● VO_2_	**0.924**
	LA ● HR	0.401
	LA ● PO	**0.910**
	LA ● VO_2_	0.753

Values in bold are statistically.

## Data Availability

All data are presented in the study.

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
