# Peer review of "The Impact of Four High-Altitude Training Camps on the Aerobic Capacity of a Short Track PyeongChang 2018 Olympian: A Case Study"

_ijerph, 2022, doi:10.3390/ijerph19073814_

Round 1

Reviewer 1 Report

Dear authors,

The present study aims to characterize the training process in high mountain conditions and its impact on the physical fitness in an elite skater specializing in short track. I believe that the case reports of elite athletes present a low scientific evidence and it´s located in  the bases of the pyramid of scientific evidence (in the top is located meta-analysis), however it´s very interesting for trainers based on the troublesome of finding information of elite athletes. In this way, I want to grant to the authors the efforts for trying publishing these data. Nevertheless, this manuscript is not suitable for being accepted in a scientific journal. Based on it, my only decision is rejecting this manuscript.

The major concerns about this manuscript are the next:

  • References used are clearly insufficient (only 21 references) and the majority of a very low scientific level (books, chapter of books, preprint and articles published in journals without impact factor).
  • Backgrounds are very poor. In this sense, authors haven´t differentiated between acute and adaptative responses to athletes exposed to hypoxia. Authors haven´t described different strategies as living low and training high, living high and training low…
  • Regarding to the previous commentary, authors have failed because they have confounded some physiological and performance variables. This is based on a very poor language uses for explaining the responses to hypoxia. For example, it´s not explained the effect of different enzymes that provokes some of the effect named along the introduction section. This is applicable to the characterization of the sport modality too.
  • The poor writing and physiological terms used in the text is reflexed in the incorrect terms as “aerobic” or “anaerobic lactic acid”. Firstly, lactic is not the same that lactic acid. Pleased, you must revise and modernize some physiology terminology and concepts. It´s not possible to include these terms currently.
  • What is the basis of the training periodization. I´m impacted about the absence of any solid principles of the periodization used in this study.
  • In a case report, how is it possible don´t include the training load registered? It´s not the same the training periodized than load training. If you don´t assess training load, you cannot present your case report because isn´t possible to compare periodized and its effect in the organism of athlete.
  • Do you consider that a cycle-ergometer test is a good indicator of performance in a skater specializing in short track? The tests used are not specific of the sport modality. In fact, in an elite skater a cycle-ergometer test could not detect differences in sport performance skating.
  • Why don´t you include the results of the Wingate test?
  • I have assessed to different Olympic athletes in the laboratory and it´s not possible to detect differences higher than 10 ml/kg/min. Under my experience, differences along the time are very low and the more important differences is based on power or velocity located to a determined VO2 or to thresholds or maximal aerobic speed or maximal aerobic power. I cannot believe that this athlete could have an VO2max of 27.4 ml/kg/min in 2018. This assessment and other are incorrect. I´m sure about it. It´s impossible to increase more than a double VO2max in a human.
  • Based on the previous important concerns, discussion and conclusion section haven´t got any confidence.

Author Response

Dear Reviewer,

Thank you very much for your time and valuable comments, which all have been considered and incorporated. The detailed list of responses is given below. We hope that the modifications and explanation will be acceptable for you.

Yours sincerely,

Rydzik, corresponding author

The present study aims to characterize the training process in high mountain conditions and its impact on the physical fitness in an elite skater specializing in short track. I believe that the case reports of elite athletes present a low scientific evidence and it´s located in  the bases of the pyramid of scientific evidence (in the top is located meta-analysis), however it´s very interesting for trainers based on the troublesome of finding information of elite athletes. In this way, I want to grant to the authors the efforts for trying publishing these data. Nevertheless, this manuscript is not suitable for being accepted in a scientific journal. Based on it, my only decision is rejecting this manuscript.

A: Thank you for your valuable comments and observations. We realize that the works describing the case study of elite athletes do not provide the possibility of drawing conclusions on a large scale of the population, but precisely because they concern elite athletes, they are cognitively valuable. In the available scientific literature, there is no work on this type of individual, let alone those mentioned by name and surname (easy to identify). Therefore, each work describing such exceptional people (constituting in total less than a per mille of humanity's population) contributes a lot to cognitive knowledge and also has a certain application aspect. But, of course, we realize that not everyone has to share these views. Contemporary training of athletes at the highest (Olympic) level is more and more often based on scientific works, evidence-based sports medicine, and thus any publication describing the relationship between the applied training and adaptive changes translating into sports results is valuable from both a scientific and practical point of view. Of course, we also realize that each case report is only (or even) a description of a specific human (individual) and cannot be translated into the general population, so we tried not to use sophisticated and artificially bent statistical techniques to prove the significance of changes. However, we wanted to note that we do not consider these studies to be of little scientific (cognitive) value. At the same time, we share the opinion of the Reviewer that the studies on the meta-analysis are more useful and credible in this respect as scientific evidence. However, assuming that case studies do not have such features, we would also have to negate the value of similar publications, e.g. in the context of rare diseases in clinical medicine (and these are generally considered as important reports).  

The major concerns about this manuscript are the next:

References used are clearly insufficient (only 21 references) and the majority of a very low scientific level (books, chapter of books, preprint and articles published in journals without impact factor).

A: Thank you for your valuable opinion, we have added literature of a higher scientific level (as suggested by the Reviewer)

Backgrounds are very poor. In this sense, authors haven´t differentiated between acute and adaptative responses to athletes exposed to hypoxia. Authors haven´t described different strategies as living low and training high, living high and training low.

A: Thank you for your valuable comments, we have added a paragraph about the LH-TL, LL-TH , LH, TH altitude training)

Regarding to the previous commentary, authors have failed because they have confounded some physiological and performance variables. This is based on a very poor language uses for explaining the responses to hypoxia. For example, it´s not explained the effect of different enzymes that provokes some of the effect named along the introduction section. This is applicable to the characterization of the sport modality too.

A:To be honest, we don't actually understand this comment. I don't know what parameters were confused.

The poor writing and physiological terms used in the text is reflexed in the incorrect terms as “aerobic” or “anaerobic lactic acid”. Firstly, lactic is not the same that lactic acid. Pleased, you must revise and modernize some physiology terminology and concepts. It´s not possible to include these terms currently.

A: The study has been edited for correct language use.

What is the basis of the training periodization. I´m impacted about the absence of any solid principles of the periodization used in this study.

A: The paper deals with a part of a one-year training cycle. Due to a high level of athletic competition, training details were not disclosed in the paper.

In a case report, how is it possible don´t include the training load registered? It´s not the same the training periodized than load training. If you don´t assess training load, you cannot present your case report because isn´t possible to compare periodized and its effect in the organism of athlete.

A: We evaluated training effects and the comparisons concerned only the effects.

Do you consider that a cycle-ergometer test is a good indicator of performance in a skater specializing in short track? The tests used are not specific of the sport modality. In fact, in an elite skater a cycle-ergometer test could not detect differences in sport performance skating.

A: Cycloergometric tests are offen used in the diagnosis of general physical fitness (both aerobic and anaerobic) of speed skaters. Moreover, many speed skaters use cycling training as one of their main general training tools.

Why don´t you include the results of the Wingate test?I have assessed to different Olympic athletes in the laboratory and it´s not possible to detect differences higher than 10 ml/kg/min. Under my experience, differences along the time are very low and the more important differences is based on power or velocity located to a determined VO2or to thresholds or maximal aerobic speed or maximal aerobic power. I cannot believe that this athlete could have an VO2max of 27.4 ml/kg/min in 2018. This assessment and other are incorrect. I´m sure about it. It´s impossible to increase more than a double VO2max in a human.

A: We agree with the Reviewer's opinion that it is impossible to double the VO2max results in such a short time, and even more so for an elite athlete. However, we would like to note that the lowest VO2max value that our athlete had was 48.3 ml / kg / min, and not 27.4 ml / kg / min, as suggested by the Reviewer. Perhaps Table 5 is not readable, but the value of 27.4 ml / kg / min refers to oxygen uptake at the intensity at the threshold of anaerobic changes (AT4), which means that in 2018 at the intensity at the AT level (lactate at the level of 4mmol / l) consumed 27.4 ml of oxygen / kg / min. These were not the maximum values (which in this test amounted to 52.9 ml / kg / min). In the period studied, our athlete's VO2max ranged from 48.3 to 61.0 ml / kg / min. On the other hand, the value of oxygen uptake (VO2) at the threshold intensity (AT4) ranged from 27.4 to 51.3 ml / kg / min  

Based on the previous important concerns, discussion and conclusion section haven´t got any confidence.

A: While respecting the Reviewer's view, unfortunately we cannot agree with this remark. All the presented results are based on reliable measurements and test results, and undoubtedly they were the result of hard training work and the applied high-altitude camps. Perhaps the current (extended and revised version of the manuscript) will convince the Reviewer to change his decision.

Reviewer 2 Report

The present study is of interest to describe a characteristic of high-altitude training camps and its influence on the aerobic capacity of the Polish representative (M.W.), participant of the PyeongChang 2018 Winter Olympic Games. 

I have some minor and one major recommendation to improve the quality of the manuscript.

(Minors):

Abstract

  1. Authors should insert some anthropometric information (age, sex, IMC,...);
  2. There are some abbreviations that need to be defined (for instance, AT-PO); 

Introduction

  1. Authors should present all hypotheses that were tested.

(Major concern)

Materials and Methods 

  1. The authors should present some inferential statistical analyses. I fully understand that as a case study the majority of the data reported is descriptive (and was very well presented). For instance, did you find any statistical difference between training camps for any of the physiological parameters that were collected? Did you find any associations between these physiological parameters? Did you find any associations between these physiological parameters when considering the Δ between training camps? In conclusion (L 117-119), the authors wrote that "2. A significant improvement in physiological parameters was found, some of them reaching the highest values in the professional skater's career so far: threshold of anaerobic changes AT-VO2 [ml/kg/min] – 51,3, and VO2max [ml/kg/min] alone 61,0." However, and if I understood well no inferential statistical analyses were used to reinforce this finding. In fact, the word "significant" it was used several times during the discussion section, however, no statistical analyses were presented to reinforce your findings. 

Overall, I truly recommend the author to explore with more detail your interesting data, using some robust analyses (i.e, inferential) to better understand your findings. 

Author Response

Dear Reviewer,

Thank you very much for your time and valuable comments, which all have been considered and incorporated. The detailed list of responses is given below. We hope that the modifications and explanation will be acceptable for you.

Yours sincerely,

Rydzik, corresponding author

The present study is of interest to describe a characteristic of high-altitude training camps and its influence on the aerobic capacity of the Polish representative (M.W.), participant of the PyeongChang 2018 Winter Olympic Games. 

I have some minor and one major recommendation to improve the quality of the manuscript.

(Minors):

Abstract

  1. Authors should insert some anthropometric information (age, sex, IMC,...);
  2. There are some abbreviations that need to be defined (for instance, AT-PO); 

A: Information has been added

Introduction

Authors should present all hypotheses that were tested.

A: The hypothesis has been added

(Major concern)

Materials and Methods 

  1. The authors should present some inferential statistical analyses. I fully understand that as a case study the majority of the data reported is descriptive (and was very well presented). For instance, did you find any statistical difference between training camps for any of the physiological parameters that were collected? Did you find any associations between these physiological parameters? Did you find any associations between these physiological parameters when considering the Δ between training camps? In conclusion (L 117-119), the authors wrote that "2. A significant improvement in physiological parameters was found, some of them reaching the highest values in the professional skater's career so far: threshold of anaerobic changes AT-VO2 [ml/kg/min] – 51,3, and VO2max [ml/kg/min] alone 61,0." However, and if I understood well no inferential statistical analyses were used to reinforce this finding. In fact, the word "significant" it was used several times during the discussion section, however, no statistical analyses were presented to reinforce your findings.

A: This has been complemented as suggested 

Overall, I truly recommend the author to explore with more detail your interesting data, using some robust analyses (i.e, inferential) to better understand your findings. 

Reviewer 3 Report

Introduction:

  • In my opinion, some information is unnecessarily divided into separate paragraphs, eg paragraphs 1 and 2 could be combined, as well as 3 and 4. However, this is just my feeling, to which the authors do not have to refer if they do not want to.
  • The second paragraph (lines 38 to 50) lacks citations from the scientific literature. This type of information must be supported by previous research on this topic.
  • Sentences on lines 48-50 and 59-61 are duplicated.
  • Paragraph 2 (lines 38-50) and paragraph 5 (lines 59-69) refer to the same topic, slightly changing the scope of the information. I don't understand why they are separated by information about short-track speed skating. Consider the order of the paragraphs. It would be best to start the introduction with a description of the discipline, and then put together all the information about high-altitude training.
  • Paragraph 7 nicely combines the two aspects discussed (high-altitude training and the short-track speed skating). And the last paragraph clearly presents the goals of the work.

Materials and Methods:

  • Invalid sequence of tables. Table 3 appears first in the text of the work. Additionally, this reference is not to the aforementioned table 3 but probably to table 5.
  • The full name, manufacturer, country of origin of the cycloergometer should appear at the first mention of it, on line 137.
  • Paragraph order: the blood lactate concentration information (lines 161-163 and 166-167) are divided by the heart rate information (lines 164-165).
  • The lines 189-201 refer to the same thing, so they could be combined into one paragraph.
  • The sentence "Table 2 below shows the training means used during the individual mesocycles and the number of training units performed (tables 3 and 4)." (lines 217-218) would be more readable as "Table 2 below shows the training means used during the individual mesocycles and tables 3 and 4 shows the number of training units performed."

Results:

  • I have no comments.

Discussion:

  • Lines 84-86 in the discussion are redundant, this information should be presented only in the material and methods chapter.
  • Information on lines 87-90 from the discussion, repeated with lines 209-212 from the material and methods section.
  • The results are described once again in the discussion. If this is the case, we should compare them with the results from other studies.
  • The discussion should be heavily rewritten.

Conclusions:

  • Clearly written conclusions.

The research seems very interesting and the topic has an important practical application. However, the form of presenting the publication should be worked on.

Author Response

Dear Reviewer,

Thank you very much for your time and valuable comments, which all have been considered and incorporated. The detailed list of responses is given below. We hope that the modifications and explanation will be acceptable for you.

Yours sincerely,

Rydzik, corresponding author

Introduction:

  • In my opinion, some information is unnecessarily divided into separate paragraphs, eg paragraphs 1 and 2 could be combined, as well as 3 and 4. However, this is just my feeling, to which the authors do not have to refer if they do not want to.

A: This part has been corrected

  • The second paragraph (lines 38 to 50) lacks citations from the scientific literature. This type of information must be supported by previous research on this topic.

Reference has been added

  • Sentences on lines 48-50 and 59-61 are duplicated.

A: Lines 48-51 have been removed

  • Paragraph 2 (lines 38-50) and paragraph 5 (lines 59-69) refer to the same topic, slightly changing the scope of the information. I don't understand why they are separated by information about short-track speed skating. Consider the order of the paragraphs. It would be best to start the introduction with a description of the discipline, and then put together all the information about high-altitude training.

A: This part has been corrected

  • Paragraph 7 nicely combines the two aspects discussed (high-altitude training and the short-track speed skating). And the last paragraph clearly presents the goals of the work.

A: Thank you very much

Materials and Methods:

  • Invalid sequence of tables. Table 3 appears first in the text of the work. Additionally, this reference is not to the aforementioned table 3 but probably to table 5.

A: This part has been corrected

  • The full name, manufacturer, country of origin of the cycloergometer should appear at the first mention of it, on line 137.

A: This part has been corrected

  • Paragraph order: the blood lactate concentration information (lines 161-163 and 166-167) are divided by the heart rate information (lines 164-165).

A: This part has been corrected

  • The lines 189-201 refer to the same thing, so they could be combined into one paragraph.

A: This part has been corrected

  • The sentence "Table 2 below shows the training means used during the individual mesocycles and the number of training units performed (tables 3 and 4)." (lines 217-218) would be more readable as "Table 2 below shows the training means used during the individual mesocycles and tables 3 and 4 shows the number of training units performed."

A: This part has been corrected

Results:

  • I have no comments.

A: Thank you very much

Discussion:

  • Lines 84-86 in the discussion are redundant, this information should be presented only in the material and methods chapter.
  • Information on lines 87-90 from the discussion, repeated with lines 209-212 from the material and methods section.
  • The results are described once again in the discussion. If this is the case, we should compare them with the results from other studies.
  • The discussion should be heavily rewritten.

A: The discussion has been rewritten

Conclusions:

  • Clearly written conclusions.

 A: Thank you very much

The research seems very interesting and the topic has an important practical application. However, the form of presenting the publication should be worked on.

Round 2

Reviewer 2 Report

The authors did a good job of reviewing the manuscript and answering the revisions made. 

However, I can not understand (if I understood well) why the authors did not present some inferential statistical analyses as proposed in the first round of revision. Across the text, the authors indicate, for instance, that:   

L 22 "significant improvements"

L 24- 26 " significant increase" 

L 236-237 "significant production"  

Results section   

L 14-16 " significant decrease"

L 24-25 "significant improvement"

L 81-82 "significant improvement"

L 107-108 "significant exercise adaptations"

L 117-119 "significant improvement"   

Overall, the word "significant" was used several times across the text, however, no statistical analyses were presented to reinforce these findings. Thus, authors should better explain all rationale to better support all the findings.   

Author Response

Dear Reviewer

Thank you for your time in reviewing our manuscript. The statistical analysis was extended to include the calculation of Slope . Slope, sometimes referred to as gradient in mathematics, is a number that measures the steepness and direction of a line, or a section of a line connecting two points, and is usually denoted by m. Generally, a line's steepness is measured by the absolute value of its slope, m. The larger the value is, the steeper the line. Given m, it is possible to determine the direction of the line that m describes based on its sign and value: A line is increasing, and goes upwards from left to right when m > 0 A line is decreasing, and goes downwards from left to right when m < 0 A line has a constant slope, and is horizontal when m = 0 A vertical line has an undefined slope, since it would result in a fraction with 0 as the denominator. Refer to the equation provided below. Additionally, Sperman's correlation was calculated. The calculated values fully complemented and confirmed the conclusions. In addition, your verbal comments were changed in the case of the game the values turned out to be irrelevant. Thank you very much, your review has significantly improved the quality of our work.

Your Sincerly, 

Rydzik- Corresponding Author